# Exploratory Study of Superparamagnetic Iron Oxide Dose Optimization in Breast Cancer Sentinel Lymph Node Identification Using a Handheld Magnetic Probe and Iron Quantitation

**DOI:** 10.3390/cancers14061409

**Published:** 2022-03-10

**Authors:** Kanae Taruno, Akihiko Kuwahata, Masaki Sekino, Takayuki Nakagawa, Tomoko Kurita, Katsutoshi Enokido, Seigo Nakamura, Hiroyuki Takei, Moriaki Kusakabe

**Affiliations:** 1Division of Breast Surgical Oncology, Department of Surgery, Showa University School of Medicine, 1-5-8 Hatanodai, Shinagawa-ku, Tokyo 142-8555, Japan; seigonak@med.showa-u.ac.jp; 2Department of Electrical Engineering and Information Systems, Graduate School of Engineering, The University of Tokyo, 7-3-1 Hongo, Bunkyo-ku, Tokyo 113-8656, Japan; akihiro.kuwahata.b1@tohoku.ac.jp (A.K.); sekino@bee.t.u-tokyo.ac.jp (M.S.); 3Department of Electronic Engineering, Graduate School of Engineering, Tohoku University, 6-6 Aramaki Aza-Aoba, Aoba-ku, Sendai 980-8579, Japan; 4Laboratory of Veterinary Surgery, Graduate School of Agricultural and Life Sciences, The University of Tokyo, 1-1-1 Yayoi, Bunkyo-ku, Tokyo 113-8657, Japan; anakaga@g.ecc.u-tokyo.ac.jp; 5Department of Breast Surgery and Oncology, Nippon Medical School Hospital, 1-1-5 Sendagi, Bunkyo-ku, Tokyo 113-8603, Japan; tomoko28@nms.ac.jp (T.K.); takei-hiroyuki@nms.ac.jp (H.T.); 6Department of Breast Surgical Oncology, Showa University School of Medicine, Fujigaoka Hospital, 1-30 Fujigaoka, Yokohama 227-8501, Japan; enotoshi@med.showa-u.ac.jp; 7Department of Medical Device, Matrix Cell Research Institute Inc., 1-3-35 Kamikashiwada, Ushiku 300-0314, Japan; amkusa@g.ecc.u-tokyo.ac.jp; 8Research Center for Food Safety, Graduate School of Agricultural and Life Sciences, The University of Tokyo, 7-3-1 Hongo, Bunkyo-ku, Tokyo 113-8654, Japan

**Keywords:** SLN, SPIO, breast cancer, iron quantitation, magnetic probe, optimal dose, RI, SLNB

## Abstract

**Simple Summary:**

Sentinel lymph node biopsy (SLNB) using super magnetic iron oxide (SPIO) and magnetic probes is expected to be a simple and safe method of detecting cancerous lymph nodes without using radioisotopes (RIs). A multicenter trial of SLNB was conducted using a handheld magnetic probe and SPIO (Rizobist^®^) and its non-inferiority with the conventional RI method. The quantity of iron in SLN was measured to examine the necessary dosage and administration method for sufficient SLN detection in the case of this test. Further, a clinical trial was conducted to determine the possibility of SLNB with a half-dose of SPIO (1.0 mL → 0.5 mL), and the resulting iron volume measured at that time was also examined. This study demonstrates that sufficient iron content reaches SLN even at an SPIO dose of 0.5 mL.

**Abstract:**

This exploratory study compared doses of ferucarbotran, a superparamagnetic iron oxide nanoparticle, in sentinel lymph nodes (SLNs) and quantified the SLN iron load by dose and localization. Eighteen females aged ≥20 years scheduled for an SLN biopsy with node-negative breast cancer were divided into two equal groups and administered either 1 mL or 0.5 mL ferucarbotran. Iron content was evaluated with a handheld magnetometer and quantification device. The average iron content was 42.8 µg (range, 1.3–95.0; 0.15% of the injected dose) and 21.9 µg (1.1–71.0; 0.16%) in the 1-mL and 0.5-mL groups, respectively (*p* = 0.131). The iron content of the closest SLN compared to the second SLN was 53.0 vs. 10.0 µg (19% of the injected dose) and 34.8 vs. 4.1 µg (11.1%) for the 1-mL and 0.5-mL groups, respectively (*p* = 0.001 for both). The magnetic field was high in both groups (average 7.30 µT and 6.00 µT in the 1-mL and 0.5-mL groups, respectively) but was not statistically significant (*p* = 0.918). The magnetic field and iron content were correlated (overall SLNs, *p* = 0.02; 1-mL, *p* = 0.014; 0.5-mL, *p* = 0.010). A 0.5-mL dose was sufficient for SLN identification. Primary and secondary SLNs could be differentiated based on iron content. Handheld magnetometers could be used to assess the SLN iron content.

## 1. Introduction

A sentinel lymph node biopsy (SLNB) is a standard procedure for breast cancer surgery. Most metastatic cells initially drain via the lymphatic system to sentinel lymph nodes (SLNs). Cancer staging and treatment depends on SLNBs [1,2,3], and lymph nodes can be identified using dyes, tracing radioisotopes (RIs), or superparamagnetic iron oxide (SPIO) nanoparticles [4,5,6,7]. We have developed a lightweight handheld magnetometer that can be used in an intraoperative setting [8]. As shown in a previous clinical trial, this magnetometer enabled the identification of SLNs in a non-inferior way, as compared to the gold standard RI [9]. In practice, this implies that SLNBs can be performed avoiding the use of RIs and RI-specialized clinical settings that can contaminate and endanger patients and staff and that can require a strict environmental setup. With the use of SPIO nanoparticles, SLNBs can be performed in clinics without RIs and controlled radiation areas and their corresponding requirements. However, there are some issues regarding the use of SPIO nanoparticles, such as residual skin pigmentation and interference with the follow-up MRI [10,11,12].

This study, which is part of the AMED-supported project, “Research and Development of Medical Devices and Systems for Future Medical Care, Research and Development Project for Minimally Invasive Cancer Treatment Devices” (2015–March 2019), aims to optimize our technique by evaluating whether half of the original dose of SPIO nanoparticles can lead to equivalent results in SLN localization in diagnosis and staging metastatic breast cancer via biopsy. We based this project on a canine study conducted at the Laboratory of Veterinary Surgery of the University of Tokyo that showed the accumulation of a sufficient amount of iron in dog SLNs 24 h after the administration of SPIO nanoparticles [13].

We also wished to quantitate the iron load in SLNs and correlate this with the dose of SPIO nanoparticles injected, the anatomical SLN position (first vs. second lymph node draining from the cancer), and the magnetic field evaluated by a handheld magnetometer. 

This is the first clinical exploratory study to compare the doses of 1 mL and 0.5 mL of ferucarbotran SPIO nanoparticles in SLNBs using a handheld magnetometer, and to quantitate the ferucarbotran iron load according to dose and anatomical distribution.

## 2. Materials and Methods

### 2.1. Patients

Eighteen participants were prospectively enrolled in a convenience series at the Department of Breast Surgery of the Showa University Hospital, according to equipment availability within the established timeframe (June 2016–March 2019). All patients provided written informed consent for this study. The inclusion criteria were as follows: females over 20 years old diagnosed with breast cancer who were scheduled for an SLNB after clinical diagnosis of node-negative breast cancer. Nine participants were administered 1.0 mL of ferucarbotran SPIO nanoparticles (Resovist, Kyowa CritiCare, Tokyo, Japan) concurrently with administration of an RI and blue dye, and another nine were administered 0.5 mL of Resovist SPIO nanoparticles, also concurrent with an RI and blue dye. The 1.0-mL trial was carried out initially, followed later by the 0.5-mL trial.

The study’s primary endpoint was to determine the amount of administered magnetic bodies that reached SLNs. The secondary endpoints were to evaluate the number of magnetic bodies required for the adequate detection of SLNs and to determine if there is sufficient iron content in SLNs after administration of 0.5 mL (22.3 mg) of the magnetic bodies compared to 1.0 mL (44.6 mg).

This study was enrolled in the UMIN Clinical Trials Registry (UMIN000031240, 9 February 2018, Showa University Medical Ethics Committee 2382 (2016)).

### 2.2. Materials

A magnetic nanoparticle detection device that uses non-linear magnetization was utilized to quantitate iron as described by Kuwahata et al. [14].

A handheld magnetic probe was used as described by Taruno et al. [9]. This probe weighs only 108 g, is battery-powered and cordless, and was certified for use in SLNBs (CE Marking Notified Body Number: 0344; EC Certificate No.: 4201663CE01). The detection of the iron component is indicated by the magnetic flux density display on the monitor and by sound, both of which can be recognized by the surgeon.

The radioactive colloid used was 99mTc (Techne Phytate Kit, FujiFilm, Toyama Kagaku, Tokyo, Japan) and was formulated with 2.9 mg of phytate acid, 2–8 mL 99mTc, and 111 MBq 99mTc.

The RI detection system used was a 14.0-mm angle Dilon Navigator (Dilon Technologies, Newport News, VA, USA).

We used ferucarbotran (Resovist; Kyowa CritiCare, Tokyo, Japan) as the magnetic tracer. The injected material contains 540 mg/mL of ferucarbotran (27.9 of mg/mL of iron), consisting of iron oxide covered with carboxydextran. The average hydrodynamic diameter of the particle is about 57 nm.

### 2.3. Procedure

Both the radioactive colloid and ferucarbotran were administered into the areolar subcutaneous tissue up to 24 h before the surgery. The surgery was performed under general anesthesia, as described by Taruno et al. [9], by two surgeons qualified as breast cancer surgery specialists with more than 100 annual surgeries each. First, blue dye (indigo carmine, 2 mL) was injected subcutaneously. Afterward, we looked for SLNs by the magnetic probe method and then by the standard colloid RI method. When Tc accumulation was found in SLNs both with the radioisotope and handheld magnetic probes, it was determined that the nodes could be detected by both methods. Lymph nodes that were found only with the supplemental radioisotope method were determined to be nodes that could not be identified by the magnetic method. Lymph nodes that were not identified by either method were excluded from the study.

Iron content in the freshly isolated lymph nodes was measured in the operating room using an iron quantification device developed at the University of Tokyo [14]. Magnetic fields generated from the ferucarbotran in the SLN were also measured using a handheld magnetometer as described elsewhere [8]. SLNs were then promptly submitted for pathological diagnosis.

### 2.4. Statistical Analysis

This is the exploratory study. The effect of different amounts of ferucarbotran was compared using the Wilcoxon sign rank test; the correlation between iron content and the magnetic field was evaluated using the Pearson’s test. P-values above 0.05 were considered statistically significant. Data were expressed as means with standard deviations (SDs) or ranges. Data analysis was conducted using JMP Pro 15 software (SAS Institute, Cary, NC, USA). Graphs were prepared on GraphPad Prism 5.00 (GraphPad Software, San Diego, CA, USA).

## 3. Results

The baseline demographic and clinical characteristics of participants are presented in Table 1.

Eighteen women participated in this study, with nine in each group (1 mL and 0.5 mL ferucarbotran). The groups were similar regarding age and body mass index. There were more perimenopausal women in the 0.5-mL group (5/9, 44% for the 1-mL group and 7/9, 78% in the 0.5-mL group).

As for histopathology, all patients in the 1-mL group had invasive carcinoma, whereas two (22%) in the 0.5-mL group had in situ ductal carcinoma (the remainder had invasive carcinomas). Most tumors were in the upper outer quadrant for both groups (four and five for 1 mL and 0.5 mL, respectively) and sized at pT1 and pT2. The tumors of both groups had similar estrogen and progesterone receptor and HER2 status. Five of the participants in the 1-mL group and seven in the 0.5-mL group underwent a total mastectomy.

Two cases of macrometastatic lymph nodes were found in 1-mL groups. There was no statistically significant difference in the amount of iron between the lymph nodes with metastasis and the first lymph node without metastasis.

Table 2 summarizes the SLN findings. On average, two SLNs were identified per patient. 

Regarding ion content (Table 2), as expected, participants injected with half the dose (0.5 mL) had approximately half the iron content (21.9 µg) in their primary SLNs compared to those injected with 1 mL of ferucarbotran (42.8 µg) (*p* = 0.131). The overall average iron content per case (including all lymph nodes recovered) was also halved. However, these differences were not statistically significant. The percentage of iron found in the first lymph node in relation to the dose injected was similar regardless of the amount of ferucarbotran injected.

The magnetic field detected from the closest SLNs to the tumor using the magnetometer was similar and, on average, irrespective of the dose (*p* = 0.918).

The relationship between the iron content of the first and second SLNs found significantly decreased from 53.0 µg to 10.0 µg (19%) in the participants injected with 1 mL of ferucarbotran and from 34.8 µg to 4.1 µg (12.1%) in those injected with 0.5 mL (*p* < 0.001 for both groups) (Figure 1a).

Figure 1b–d show the correlation of magnetic fields of SLNs recorded using the magnetometer and the iron content measured using the iron quantification device. The correlation coefficient was over 0.5, and the linearity was significant for overall SLNs (Figure 1b) and in SLNs divided by groups according to ferucarbotran dose (Figure 1c,d).

The magnetic method was able to detect SLNs in all cases in which iron content was measured.

There were no significant side effects reported in this study.

## 4. Discussion

This study aimed to evaluate the effect of different doses of ferucarbotran injected into the breast on the iron content and on the identification of SLNs. These SLNs were probed using a magnetic handheld device developed by the Department of Electrical Engineering and Information Systems at the University of Tokyo [8]. This probe has also been applied to surgery for non-palpable breast lesions [15].

The overall iron content of the SLNs was proportional to the quantity injected (average per case of 52.4 µg, or 0.18%, of the iron injected in participants injected with 1 mL of ferucarbotran; 24.6 µg, or 0.17%, of the iron injected in participants injected with 0.5 mL of ferucarbotran), but this difference was not statistically significant. A mouse study in which there was subcutaneous injection with several amounts of Endorem SPIO nanoparticles (Guerbet, Villepinte, France) also showed no correlation with iron retention in the lymph nodes [16].

Using a handheld magnetometer, we found that the magnetic fields of the SLNs of participants injected with different doses were similar. The magnetic fields of SLNs in the overall participants and in each group were linearly correlated with their iron content. These data corroborated a previous canine study [13] carried out with ferucarbotran and one with porcine models with another SPIO nanoparticle (Sienna+; Sysmex, Kobe, Japan) [17,18]. The canine study [13] was a preclinical study carried out in spontaneous tumor-bearing dogs. Ferucarbotran was shown to be non-inferior to dye in detection of SLNs on veterinary clinical cases. The authors suggested that this SPIO nanoparticle would also be useful in human patients, even though they could not analyze RI data because of limitations in veterinary medicine regulations. These porcine and canine studies indicate that magnetometers can be used for the quantitative assessment of iron content in vivo in the clinic and that a 0.5-mL dose of ferucarbotran is sufficient for this purpose. However, our findings contradict a study performed in a porcine model [17], where, although SLN iron content was proportional to the SPIO nanoparticle dose injected, there was no association between the injected dose and the magnetic field measured with a magnetometer. However, the nanoparticle type and procedure differed from those described in our study.

A 1-mL dose of ferucarbotran led to the identification of a mean 2.2 SLNs per case and a mean 2.1 SLNs per case in those injected with 0.5 mL. The number of SLNs detected in this study are compatible with findings using other SLN identification techniques, such as the use of dyes, radioisotopes, or near-infrared probes [19].

Hersi et al. [20] also carried out a dose and timing optimization study for breast cancer SLN identification, using Magtrace (Mammotome, Cincinnati, OH, USA) (11.2 mg Fe/mL; in comparison, ferucarbotran used in our study is at 27.93 mg Fe/mL) as the magnetic tracer. They did not quantitate iron incorporation, but they also found that lower doses (1.0 and 1.5 mL, as compared to standard 2.0 mL) of magnetic tracer can be equally effective in SLN identification. They found that higher efficiency was correlated with a longer time from tracer injection to surgery, to which we also obtained similar results in mouse experiments with ferucarbotran [6]. Another dose optimization study was carried out using Sienna+ (27 mg Fe/mL) at 1.0, 1.5, and 2 mL [7], which also showed similar efficiency with lower doses, which correlated with lower skin pigmentation, indicating the benefit of using lower doses. The comparison of Magtrace and Sienna+ with ferucarbotran may be done carefully given that these vary not only in iron content, but also in nanoparticle conjugation and size, which has pharmacokinetic implications [21,22,23].

Apart from our previous study (a conference paper focusing on iron quantitation with the 1 mL ferucarbotran data), the quantification of iron content in human breast cancer patient SLNs using ferucarbotran as a tracer is new. Indeed, to our knowledge, there have been no other reports of iron content in SLNs of patients treated with any magnetic tracer. Iron content in human samples has only been previously analyzed by histological and imaging approaches [17,24,25].

We were interested in determining the correlation of the amount of iron retained in SLNs and their proximity to the tumor. Mammary SLNBs are important in breast cancer staging [26,27,28], but most clinical studies do not discuss differences in SLNs according to their position in the lymph node chain. We found that, for both doses of ferucarbotran injected, there was a significant decrease in the iron content and corresponding magnetic field from the first to the second SLN. This indicates that iron quantification can be used in the clinic to estimate whether an SLN is primary or secondary, and it can aid in ranking SLNs, which is important in diagnosis and prognosis. A rat study with ultra-small SPIO nanoparticles also detected a decrease in iron uptake in the second SLN compared to the first [29].

Although we did not detail the data analysis, we found that there was an increase in the amount of iron in SLNs with increasing times between ferucarbotran injection and surgery. Similar findings were found in a mouse study with Resovist [30] and a porcine study [18] with another SPIO nanoparticle. In this study, a biopsy was possible with the administration one day before. In comparison, Sienna+ may be used in patients 3 h to many weeks before surgery [31].

We did not evaluate skin pigmentation in our study but believe that this side effect probably decreases in patients receiving the lower dose of ferucarbotran (0.5 mL), in accordance with a study with the Sienna XP tracer [11] which found lower pigmentation when using 1 mL vs. 1.5–2 mL of the tracer, one and six months after surgery.

Among other study limitations, we would like to mention that the sample size was small. Appropriate blinding was not possible due to the experimental setup, especially iron quantitation equipment. MRI follow-up was not possible due to funding limitations. 

## 5. Conclusions

Our results imply that ferucarbotran at a dose of 0.5 mL can be used to identify SLNs with a similar efficiency to a dose of 1 mL. We found that roughly 18% of the initial dose is found in the patient’s SLNs regardless of dose. The rate of SLN identification for both doses was equivalent to that using a radioisotope tracer and blue dye. The possibility of using a lower dose of ferucarbotran may decrease drawback events such as residual pigmentation and interference with a follow-up MRI, though these effects were not evaluated in this study. Our data also indicate that a handheld magnetometer can be used for SLN localization and can distinguish primary from secondary SLNs, and they indicate that magnetic field readings correlate with SNL iron content. Overall, ferucarbotran use associated with the handheld magnetometer can substitute radioisotope tracers in SLN localization and is therefore safer for patients and staff; furthermore, it allows procedures to be carried out in operating centers without the need of RI facilities.

## Figures and Tables

**Figure 1 cancers-14-01409-f001:**
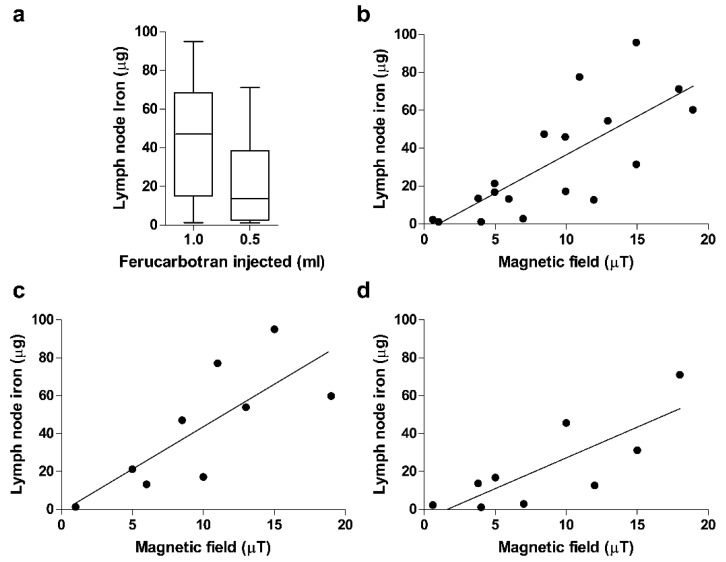
Comparison of iron content in the closest sentinel lymph node (SLN) to the tumor, according to injection volume. Biopsied SLNs had their magnetic field quantified using a magnetometer and their iron content quantified using the iron quantification device described in Kuwahata et al. [11]. (**a**). Boxplot of iron content of SLNs according to the volume of ferucarbotran injected. Pearson’s test was used to compare patients who received 1.0 mL of ferucarbotran (average iron content 42.8 μg) vs. 0.5 mL (average iron content 21.9 μg) (*p* = 0.131). (**b**–**d**) Correlation of magnetic fields recorded using the magnetometer vs. the iron content measured using the iron quantification device. (**b**) Total participants and the correlation coefficient 0.57; *p* = 0.02. (**c**) Participants who received 1.0 mL of ferucarbotran and the correlation coefficient 0.59; *p* = 0.014. (**d**) Participants who received 0.5 mL of ferucarbotran and the correlation coefficient 0.63; *p* = 0.0102.

**Table 1 cancers-14-01409-t001:** Patient and pathological data.

Characteristic	Ferucarbotran Injected		Total
	1 mL	0.5 mL	
N	9	9	18
Age (median [range]) (years)	54 (43–78)	66 (44–84)	59 (43–84)
BMI (median [range]) (kg/m)	21.3 (19.4–29.1)	20.3 (17.9–26.3)	20.9 (17.9–29.1)
Menopausal state (n)			
Premenopausal	4	2	6
Perimenopausal	5	7	12
Carcinoma type (n)			
Invasive carcinoma	9	7	16
Ductal carcinoma in situ	0	2	2
Tumor location (n)			
Upper outer quadrant	4	5	9
Upper inner quadrant	1	2	3
Lower inner quadrant	1	1	2
Lower outer quadrant	1	0	1
Central	1	1	2
Surgery type (n)			
Total mastectomy	5	7	12
Partial mastectomy	4	2	6
Pathological tumor size (n)			
pTis	0	2	2
pT1	6	4	10
pT2	2	3	5
pT3	1	0	1
Pathological lymph node status (n)			
pN0	5	9	
pNi+	0	0	
pN1mi	2	0	
pN1a	2	0	
Cases with metastatic lymph nodes/number of macrometastatic lymph nodes (percentage)	2 cases/2 nodes (10.0%)		
Hormone receptor status (n)			
Estrogen receptor (ER)+	8	7	
Progesterone receptor (PR)+	7	7	
HER2 positive	1	0	

(n): number of cases. BMI: body mass index.

**Table 2 cancers-14-01409-t002:** Sentinel lymph node (SLN) findings.

Ferucarbotran dose in volume	1 mL	0.5 mL	*p* Value
Ferucarbotran dose in mg ^a^	44.6	22.3	
Number of participants	9	9	
Time from administration to surgery (median, range)	20 h (5–24)	20 h (20–24)	
Number of isolated lymph nodes (case average)	20 (2.2)	19 (2.1)	
Iron content (µg)			
Average (range; percentage^c^) iron content in 1st SLN (µg)	42.8 (1.3–95.0, 0.15%)	21.9 (1.1–71.0, 0.16%)	0.131
Average iron content per case ^b^ (percentage ^c^) (µg)	52.4 (0.17–94.9) (0.18%)	24.6 (0.7–71) (0.17%)	0.073
Iron content of first and second SLN (percentage ^d^) (µg)	42.8(1.32–94.9) vs. 10.0 (4.02–27.5) (19%)	21.9 (1.1–71) vs. 4.1 (0.7–11.2) (11.8%)	0.001/0.001 ^e^
Iron content of macrometastatic LN and non-metastatic first SLN(µg)(range)	45.2 (13.2–77.0)-vs. 50.4 (1.32–94.9)		0.95
Average (range) ion magnetic field in 1st SLN (µT)	7.3 (1.0–19.0)	6.0 (0.5–18.0)	0.918

^a^ As reference, iron content of ferucarbotran = 27.93 mg/mL. ^b^ Includes all lymph nodes recovered per patient; ^c^ average percentage of the administered amount; ^d^ average percentage of iron found in the second SLN as compared to the first; ^e^ comparison of first SLN vs. second SLN per dose~approximately.

## Data Availability

The datasets generated during and/or analyzed during the current study are not publicly available to protect patient confidentiality but are available from the corresponding author on reasonable request.

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
