# Peer review of "Exploratory Study of Superparamagnetic Iron Oxide Dose Optimization in Breast Cancer Sentinel Lymph Node Identification Using a Handheld Magnetic Probe and Iron Quantitation"

_cancers, 2022, doi:10.3390/cancers14061409_

Round 1
Reviewer 1 Report
Several of my previous comments have been addressed, but not very satisfactorily. I think that the manuscript needs a more extensive revision than the small changes performed.
The idea is novel and interesting but needs support with more sound structure. It is important to get this data out, but more discussion in relation with the SPIO literature will support the manuscript
Author Response
I would like to thank you for your proper advice.
I think this study needs to be redesigned, as you pointed out. This is an exploratory study and currently it is not possible to add cases. Based on this paper, we would like to conduct a large-scale study.
Reviewer 2 Report
I am satisfied with the changes made.
Author Response
I thank you for your proper advice.
This manuscript is a resubmission of an earlier submission. The following is a list of the peer review reports and author responses from that submission.
Round 1
Reviewer 1 Report
Thank you for letting me review this manuscript. As mentioned it is a small study looking at iron content in sentinel lymph nodes after two different doses of ferucarbotran. I have som minor comments
The penultimate sentence in the simple summary needs to be clarified.
In the simple summary you call te nanoparticles magnetic and in the abstract superparamagnetic.
In line 41 in the abstract you state that "it" was nor significant. What was not significant?
Table 1 need to be clarified regarding lymph node status. I woud include all 43 removed nodes.
HER2 status should be HER2 positive I presume.
Results line 156 does not match table 1.
Line 160-163. To me it became unclear if you found these node withe the magnetometer or not. If not, this must be clear in the discussion as well. THe mean number of nodes found by the different methodes should be presented.
Discussion line 247. The reference is wrong I believe. THis was not a dose optimising study.
Author Response
Dear. Reviewer
We are very grateful for your careful and accurate peer review. Thank you for your suggestion.
Point1: The penultimate sentence in the simple summary needs to be clarified.
Respons1: Thank you for your advice. I'm very sorry for the incomprehensible. I changed “We also measured the iron content of SLN at the dose of 0.5 ml.”
Point2: In the simple summary you call te nanoparticles magnetic and in the abstract superparamagnetic.
Respons1: Thank you for pointing out my mistake. I changed from magnetic to SPIO, Line23 and 26.
Point3: In line 41 in the abstract you state that "it" was nor significant. What was not significant?
Respons3: It means that there is no statistically significant difference. I attached “statistical” sfter “significant”.
Point4: Table 1 need to be clarified regarding lymph node status. I woud include all 43 removed nodes.
Respons4: All cases scheduled for SLNB are judged to have no metastasis before surgery.
The judgment of lymph nodes after surgery is described in the pathological LN status of table1. No metastases except those with metastases.
Point5:HER2 status should be HER2 positive I presume.
Response5: Thank you for pointing out my mistake. I changed HER2 positive.
Point6: Results line 156 does not match table 1.
Respons6: All cases in the 1 ml administration group had invasive cancer. The 0.5 ml group contained two cases of non-invasive cancer. Table 1 shows that
Point7: Line 160-163. To me it became unclear if you found these node withe the magnetometer or not. If not, this must be clear in the discussion as well. THe mean number of nodes found by the different methodes should be presented.
Respons7:Thank you for your advice. This is not very meaningful and confusing, so I deleted it.
Point8: Discussion line 247. The reference is wrong I believe. THis was not a dose optimising study.
Respos8: Thank you for pointing out. I refer to the number of SLNs mentioned in this article.
I changed line245-247.
The number of SLNs detected in this study. These values are compatible with findings with other SLN identification techniques, such as the use of dyes, radioisotopes, or near-infrared probes.
Again, thank you for giving us the opportunity to strengthen our manuscript with your valuable comments and queries. We have worked hard to incorporate your feedback and hope that these revisions persuade you to accept our submission.
Sincerely,
Kanae Taruno

Reviewer 2 Report
The authors test a reduced iron dose for SPIO as a SLN tracer for breast cancer. They use two groups of 9 patients each for this study and conclude that 0.5 ml of ferucarbotran is as efficient as 1 and that iron concentration in the "second nodes" was much lower for each dose.
On the other hand, the second nodes are hard to find with the magnetometer in the 0.5 group.
Some things to improve the study:
Please go with medians and range.
Provide the exact time of injection between groups.
Provide the study power and sample size calculation
In the Discussion:
The study has too few participants to make any claims on the clinical outcomes.
Please shorten; it is overlong.
Author Response
Response to Reviewer 1 Comments
Dear. Reviewer
We are very grateful for your careful and accurate peer review. Thank you for your suggestion.
Point1: Please go with medians and range.
Response1:Thank you for your advice. I added it to Table 2
Point2: Provide the exact time of injection between groups.
Respons2: Thank you for your advice. I also added it to Table 2
Point3: Provide the study power and sample size calculation
Resposs3: You have raised an important point; however, I am sorry it is difficult. Since this study was conducted exploratory, the number of cases is very small. Therefore, it is not possible to add cases and it is difficult to prove that the number of cases is sufficient.
In the Discussion:
The study has too few participants to make any claims on the clinical outcomes.
Point4:Please shorten; it is overlong.
Respons4: Thank you for your advice. Certainly the sample size of this study is small. I changed the text of the discussion short. I delete Line 285-297.
Again, thank you for giving us the opportunity to strengthen our manuscript with your valuable comments and queries. We have worked hard to incorporate your feedback and hope that these revisions persuade you to accept our submission.
Sincerely,
Kanae Taruno

Round 2
Reviewer 2 Report
In the revised version of the manuscript, the authors provide minor changes.
There is still the methodological choice of mean and range (Age and BMI, Table 1), which is not correct.
The Discussion is still overlong and the authors make the same claims. Methodologically, it is not appropriate to claim that "it works" in a "convenience series" of 9+9 which means that selection bias is very high. The sample size is too small to overcome and the authors' reply that it is just a limitation (row 303-304) is insufficient. And this is a limitation impossible to overcome if the authors choose to claim in the beginning of the Discussion that "This study aimed to evaluate the effect of different doses of ferucarbotran injected into the breast on the iron content and on the identification of SLNs". If the authors chose to describe only the amount of iron on humans by a dose of 0.5 ml or equivalent, omitting the discussion about SLN detection and the discussion that it is feasible, then that would be more solid. Not perfect, but at least solid. Then we would be discussing about a small case series, at best a phase 1 trial, but methodologically more appropriate.
Then the entire Discussion has unclarities, inconsistencies and inaccuracies. Some examples:
- the authors compare the probe they used with the UK probe (Sentimag) making claims on advantages of their probe, but there are no studies comparing the probes, so these claims are arbitrary.
- Then they claim that the number of SLNs retrieved is the same regardless of dose, but higher up, in the results, they show that the 0.5 ml dose might not have found the "second" SLN. The fact that these outcomes did not reach statistic significance obviously highlights the problem of severely underpowered studies.
- Then, when discussing the study findings by Hersi, they comment on a time parameter (the injection in the preoperative setting), they state that they found the same thing but do not show the data. They need to show the data.
- "Sienna+ can be injected 3-15 hours before surgery" (row 284-285) whereas they then cite a study by Hersi where authors injected up to 7 days and there is the SentiNot study from the Swedish group where it is injected many weeks before.
- They write that Sienna+ and MAgtrace do not only differ in iron concentration but in other properties that "may" affect outcomes. Then they cite three articles that do not support the claim.
- Then, in the Sunrise trial, by Rubio et al, there is statistical difference in discoloration between doses only for 1 ml compared to 1.5-2 ml, but the authors here make the claim that the difference is between 1-1.5 and 2 (rows 301-302).
In conclusion, the manuscript revision did not succeed in meeting the mark overall.